# Action execution and action observation elicit mirror responses with the same temporal profile in human SII

Maria Del Vecchio[1,2 ✉], Fausto Caruana[2], Ivana Sartori[3], Veronica Pelliccia[3], Flavia Maria Zauli[4], Giorgio Lo Russo[3], Giacomo Rizzolatti[2,5] & Pietro Avanzini[2]

The properties of the secondary somatosensory area (SII) have been described by many studies in monkeys and humans. Recent studies on monkeys, however, showed that beyond somatosensory stimuli, SII responds to a wider number of stimuli, a finding requiring a revision that human SII is purely sensorimotor. By recording cortical activity with stereotactic electroencephalography (stereo-EEG), we examined the properties of SI and SII in response to a motor task requiring reaching, grasping and manipulation, as well as the observation of the same actions. Furthermore, we functionally characterized this area with a set of clinical tests, including tactile, acoustical, and visual stimuli. The results showed that only SII activates both during execution and observation with a common temporal profile, whereas SI response were limited to execution. Together with their peculiar response to tactile stimuli, we conclude that the role of SII is pivotal also in the observation of actions involving haptic control.

[1] University of Modena and Reggio Emilia, Dipartimento di Scienze Biomediche, Metaboliche e Neuroscienze, 41100 Modena, Italy. [2] Istituto di Neuroscienze, Consiglio Nazionale delle Ricerche, 43125 Parma, Italy. [3] Centro per la Chirurgia dell'Epilessia "Claudio Munari", Ospedale Ca' Granda—Niguarda, 20162 Milano, Italy. [4] Università degli Studi di Milano, Dipartimento di Scienze Biomediche e Cliniche "L. Sacco", 20157 Milano, Italy. [5] University of Parma, Dipartimento di Medicina e Chirurgia, 43125 Parma, Italy. ✉email: maria.delvecchio@unimore.it

                                                                                                      1

It is well accepted that the secondary somatosensory area (SII) plays a fundamental role in processing tactile inputs[1], making it a high-order hub for somatosensory processing[2]. It has bilateral receptive fields with a somatotopic organization[3–8] and receives projections mainly from the thalamus[9] and from primary somatosensory area SI[10,11]. Compared to SI during the processing of tactile inputs, SII exhibits longer and more complex responses[12,13], possibly underlying high-order functions, such as object identification, tactile learning and memory[14–21].

From a cytoarchitectonical point of view, classical studies indicated that SII is part of the inferior parietal lobule (IPL), and corresponds to portions of Brodmann areas 40 and 43[22,23]. Recent parcellations[24] excluded SII from the inferior parietal lobule (IPL), limiting it to seven different areas covering area 39 on the angular gyrus (PGa and PGp) and area 40 on the supramarginal gyrus up into the Sylvian fissure (PFop, PFt, PF, PFm, PFcm). A study specifically devoted to the human parietal operculum (OP) identified four different architectonic subdivisions, i.e., OP1-4[25], and demonstrated that OP1, the most caudal area, represents the homolog of Macaque SII. In line with this view, OP1 presents the strongest activation following somatosensory stimulation among all opercular areas[26]. This fine-grained parcellation led to the assumption that SII and IPL are functionally distinct, with visuo-motor responses limited to the latter.

Recently, single-neurons studies in monkeys showed that beyond tactile inputs, SII responds to a wide number of stimuli, including peri-personal space stimulation, active hand movements, proprioception, observation of objects displacement, and observation of reaching and grasping actions[27–29]. These data are in agreement with both monkey and human functional magnetic resonance imaging (fMRI) studies, reporting SII activation during the observation of another individual's body being touched[30–33]. Taken together, these results (a) suggest that the notion of SII as a purely somatosensory area should be revised, and (b) highlight a possible continuity of SII within IPL, embedding SII also in visuo-motor transformation[34,35].

To better address these issues, we investigate the response of bilateral SII in a motor task requiring action planning, object reaching, grasping and manipulation. We then compared all these responses with those recorded during the observation of the same actions performed by another individual. Since the processing of somatosensory inputs is contralateral, we extended the study to left SI (3a, 3b, 1), which is known to process both tactile (3b, 1) and proprioceptive inputs (3a)[19]. Although connections with SII are reciprocal, projections from SI to SII are more important and represent the main input for this latter[26,36,37].

The present study has been conducted recording stereotactic electroencephalography (stereo-EEG) in patients affected by drug-resistant epilepsy, whose implantation covered bilateral SII or contralateral SI. While several neuroimaging studies tackled cortical activity sustaining active movements, executing actions, e.g., within a scanner sets a number of constraints (lying position, limited space exploration, altered visual feedback) limiting the implementation of a truly ecological behavior. Electrophysiological recordings[38] have then been chosen to overcome these limitations, however, their poor localization power, their limited spectral content, as well as their sensitivity to movement artifacts impeded to obtain a detailed description of the neural activity sustaining active and ecological complex movements. Intracranial recordings virtually bridge the advantages of these two approaches. By combining both high-temporal resolution and localization power[39,40], this technique is the ideal tool to investigate not only the neural correlates sustaining full-fledged actions, but also the neural dynamics underlying single motor acts. Within stereo-EEG signal, we selected gamma-band activity modulation as indicator of neural activation since it is agreed to reflect spiking activity[41]. Furthermore, it is reported that this indicator is highly functionally and spatially specific in several studies[42,43].

Given these premises, the aim of the present study was to elucidate the temporal dynamics exhibited by SII and SI during the execution and the observation of complex actions. Our results show that SII activates bilaterally both during action execution and observation and, for the first time, that these conditions share an identical temporal profile. Since we found a sustained activity during the manipulation phase in both conditions, we speculate that SII is endowed with the mirror mechanism encoding the haptic component of an action. The absence of activation of SI in the observation condition suggests that SII responses are sustained by a neural circuit, able to operate simultaneously and independently from the somatosensory input.

## Results

**Sampling**. The data were analyzed in all patients whose implantation, co-registered on a template, included electrodes exploring SII or left SI (3a, 3b, 1). Overall, 18 patients were considered, 4 implanted bilaterally, 6 only in the left hemisphere, and 8 only in the right one.

SII (but not SI) was explored in 14 patients (five left, eight right, and one bilaterally), two patients' implantation covered left SI (but not SII), while other two patients showed implantations exploring both SII (bilaterally) and left SI. As far as individual leads are concerned, 63 leads recorded activity from SII (31 left, 32 right), while 12 from left SI.

**Reactivity of SII**. For each lead, the gamma power (55–145 Hz) activity was measured for each phase of the action separately, both for execution and observation conditions, and its significance tested against the 300 ms long baseline (−350, −50 ms). The results are summarized in Table 1.

In the execution phase, about half of the SII sites showed a significant gamma power increase during both the reach-to-grasp (18 left, 16 right) and the manipulation phase (19 left, 16 right) while only 3 leads were responsive in the preparation phase (2 left, 1 right). Most of the leads active during reach-to-grasp were active also during manipulation (15 left, 15 right).

When the task was performed by the experimenter (observation condition), 19 (7 left, 12 right) leads showed a significant gamma power increase during the reach-to-grasp phase, and 13 (6 left, 7 right) did the same for the manipulation phase. Overall

**Table 1** Table 1 indicates for each phase of the experimental paradigm the number of SII leads responsive only in execution (first row), both in execution and in the observation condition (second row) and only in the observation (third row) separately for contralateral and ipsilateral hemisphere.

| SIDE/PHASE | Preparation | Reach to Grasp | Manipulation |
|---|---|---|---|
| Left (contra) | 2 (exe) | 11 (exe) | 13 (exe) |
|  | 0 (exe-obs) | 7 (exe-obs) | 6 (exe-obs) |
|  | 0 (obs) | 0 (obs) | 0 (obs) |
| Right (ipsi) | 1 (exe) | 5 (exe) | 9 (exe) |
|  | 0 (exe-obs) | 11 (exe-obs) | 7 (exe-obs) |
|  | 0 (obs) | 1 (obs) | 0 (obs) |

As a remark, virtually no leads (only one in reach-to-grasp phase) were responsive only in observation condition.
Contra contralateral, ipsi ipsilateral, exe execution, obs observation.

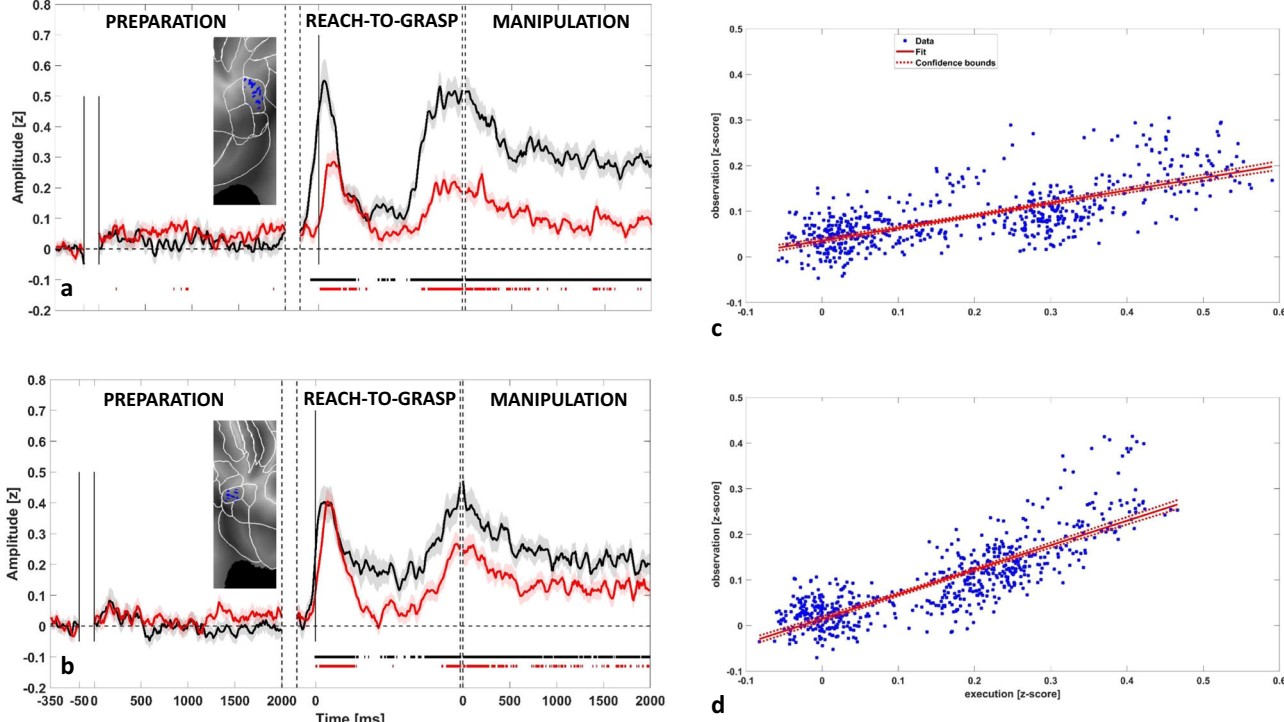

**Fig. 1 Contralateral and ipsilateral SII gamma responses to action execution and observation.** The figure depicts for the contralateral SII (**a**) and ipsilateral SII (**b**) the average time-course (±SE) for execution (black trace) and observation condition (red trace). The averaged amplitude is computed in terms of z-score respect to the baseline for each trial (see Methods) including all leads responsive in at least one phase (22 left, 19 right). Significance respect to the baseline is shown below the traces in the same color code. For visualization purposes, data are Savitzky–Golay filtered. For each hemisphere, the panel includes an inset showing the localization of responsive leads on a flat map (see Supplementary Information). **c**, **d** report, respectively, for contralateral and ipsilateral SII, the distribution of z-scored amplitudes, as well as the linear regression model ($r^2 = 0.513$ for left SII and $r^2 = 0.695$ for right SII).

nine leads were active in both phases (5 left, 4 right). No leads were responsive during the preparation phase.

It is worth noting that the large majority (96%) of the leads active during action observation were also active during action execution, thus ruling out a visual reactivity independent of motor properties, but rather suggesting a link between visual and sensorimotor functions of SII. To further address this issue, a key element is the study of the time-course of SII activity during action execution and observation, where the possible parallelism between the temporal patterns would be suggestive of a mirror-like property of SII.

Figure 1 shows the gamma-band time-course for both execution (black trace) and observation (red trace) for left SII (panel a) and right SII (panel b). Curves were computed by averaging all the leads active in at least one phase, for both execution and observation. For each condition, significance against the baseline (see Methods) is shown below the traces in corresponding color code. During action execution, the first significant activation in left SII (Fig. 1a) was observed before hand lifting. This activation was followed by a marked activity decrease during reaching, and by a strong power increase preceding the hand-object interaction, which continued for the whole manipulation phase. The activity pattern during action observation was highly similar to that observed during action execution (see Fig. 1c, d), but with a lower amplitude of the activations during hand-object interaction and manipulation phase (see Supplementary Fig. 3a) and a response onset following the hand lifting (delay of 50 ms relative to action execution in terms of earliest significance for leads active both during execution and observation conditions). A detailed list of relative delays for each leads responsive both in execution and observation condition (reach-

**Table 2 Table 2 indicates for leads responsive in reaching-to-grasp both in execution and observation condition (seven left, ten right) the delay computed in terms of earliest significance after the hand lifting (difference between observation and execution), and the maximum amplitude (z-score) for each condition.**

| Left | | Right | |
|---|---|---|---|
| | **Delay after hand lifting** | | **Delay after hand lifting** |
| Lead 1 | 80 ms | Lead 1 | 40 ms |
| Lead 2 | 130 ms | Lead 2 | 50 ms |
| Lead 3 | 50 ms | Lead 3 | 20 ms |
| Lead 4 | 60 ms | Lead 4 | 30 ms |
| Lead 5 | 50 ms | Lead 5 | -10 ms |
| Lead 6 | 50 ms | Lead 6 | 20 ms |
| Lead 7 | 20 ms | Lead 7 | 120 ms |
| | | Lead 8 | 90 ms |
| | | Lead 9 | 50 ms |
| | | Lead 10 | 60 ms |

A t-test was conducted to assess the significance of the delay against the zero ($p < 0.001$).

to-grasp phase) is presented in Table 2. The delay distribution is significant against the zero ($p < 0.001$).

The biphasic temporal pattern is present also in right SII during both action execution and observation (Fig. 1b). However, virtually no differences in amplitude were observed if comparing action execution and observation (see Supplementary Fig. 3b).

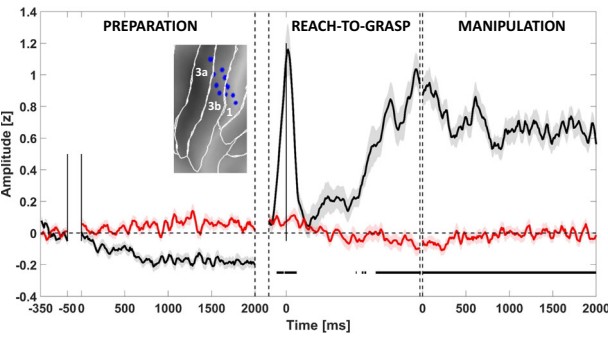

**Fig. 2 Contralateral SI gamma responses to action execution and observation.** The figure depicts for the contralateral SI (including area 3a, 3b, 1) the average time-course (±SE) for execution (black trace) and observation condition (red trace). The average is computed including all leads responsive in at least one phase (12). For visualization purposes, data are Savitzky–Golay filtered. Significant increase respect to the baseline is shown below the traces in the same color code. The panel includes an inset showing the localization of responsive leads on a flat map (see Supplementary material). It is worth nothing that no leads exploring SI complex respond to action observation.

**Table 3 Table 3 summarizes results of clinical and neurophysiological tests administered to patients.**

| Stimulation | Percentage respect to leads responsive in at least one phase | | Percentage respect all leads exploring SII | |
|---|---|---|---|---|
| | Left | Right | Left | Right |
| Contralateral median nerve stim. | 95% (21/22) | 74% (14/19) | 74% (23/31) | 64% (21/32) |
| Contralateral acoustic stim. (40 DB) | 27% (6/22) | 0% (0/19) | 20% (6/31) | 0% (0/32) |
| Contralateral acoustic stim. (85 DB) | 27% (6/22) | 26% (5/19) | 23% (7/31) | 22% (7/32) |
| Visual stimulation | 0% (0/22) | 0% (0/19) | 0% (0/31) | 6% (2/32) |
| Optokinetic stimulation | 0% (0/22) | 0% (0/19) | 0% (0/31) | 0% (0/32) |

For each stimulation percentage respect to leads responsive in at least one phase of the paradigm are reported in column 1. Column 2 reports the percentage of responsiveness respect to all leads exploring SII.

**Reactivity of SI**. Figure 2 shows the time-course of left SI activity during action execution and observation. The analyzed leads all responsive during reach-to-grasp and manipulation phases, exhibit a phasic response at the hand lifting preceded by a decrease of gamma power respect to the baseline period, covering most of the preparation phase. Before the hand-object contact and during manipulation, activity was continuously sustained.

It is important to note that both left and right SI show no activation during the observation, excluding a mirror function for this area.

Finally, the contralateral SI activation before the hand lifting, in common with SII during the execution condition, might reflect either a proprioceptive input or a tactile-off signal[44]; however, the source of this same SII activation during action observation remains unclear.

**Clinical and neurophysiological tests**. All patients examined in this study completed a stimulation set aimed at depicting the responsiveness to clinical tests including contralateral median nerve, acoustic (40 dB and 85 dB), and bilateral visual stimulation (both static and dynamic). Percentage of responsiveness respect to leads active in at least one phase of the paradigm and to all leads exploring SII are reported in Table 3.

The most relevant information emerging from this analysis is that no lead exploring SII, which is responsive to action execution or observation is also responsive to visual stimulation, even though the stimulus is suggestive of a motion, like in the optokinetic stimulation, excluding that a visual stimulus is able to activate this area. Figure 3 shows the gamma band temporal course of leads responsive to contralateral median nerve stimulation in left SI (12 out of 12 sampled, panel a) and of both left (23 out of 31) and right SII (21 out of 33) in response to median nerve stimulation (panel b). These two areas show a completely different temporal behavior: SI depicts a phasic time-course while bilateral SII has a long-lasting tonic behavior ending after 200 ms after the stimulus. Average gamma band time-courses of leads responsive to action execution and observation are shown in Supplementary Fig. 1 following contralateral (panel a), visual (panel b), optokinetic (panel b) and acoustical stimulations at 40 dB (panel d) and at 85 dB SPL (panel e).

## Discussion

In the present paper, we investigated the neural activity during the execution and the observation of reach-to-grasp and manipulation actions performed in an ecological setting. Gamma band activity was estimated from leads exploring two regions of the lateral grasping network[45], i.e., primary (SI) and secondary (SII) somatosensory areas, from 18 surgical patients. Of these, only four patients' implant sampled SI, representing a limitation for the study.

We evaluated the activity of contralateral SI and bilateral SII in three distinct action phases: motor planning, reach to grasp, and object manipulation. To complement the functional picture of SI and SII, we separately assessed their responsiveness to somatosensory (contralateral median nerve), visual (i.e., flash and optokinetic) and auditory (click) stimulations.

During action execution, SI and SII present an overlapping temporal profile (see Supplementary Fig. 4). Initially, both areas failed to show a gamma power increase during the motor planning. While this is expected as far as SI is concerned, the lack of activation in SII is less trivial, considering that this area is anatomically connected with dorso-lateral prefrontal (DLPFC) and pre-supplementary motor areas, known to be recruited during action preparation[11,46–48].

The gamma power of both SI and SII during the reaching phase is observed only at the action onset and just before the hand-object interaction, but it is virtually absent during arm movement, hence failing to reveal a contribution specific for reaching. This result is in line with a long-standing model assessing that the circuits controlling reaching movements are located more dorsally relatively to SII[49].

The most interesting result concerns the hand-object interaction, which shows a very strong gamma power response in both SI and SII. While SI activity is compatible with its tactile or proprioceptive functions, the activation of SII is also compatible, in principle, with a motor control. This latter aspect has been often neglected as SII does not respond to unspecific movement execution, but rather it activates upon a specific class of actions implying manipulation, and more generally haptic functions. The notion that SII plays a pivotal role in high-level haptic perception has been advanced on the basis of previous imaging studies in humans[50–52]. In a study on post-stroke patients, Forss et al.[53] concluded that fine manual control and haptic perception closely

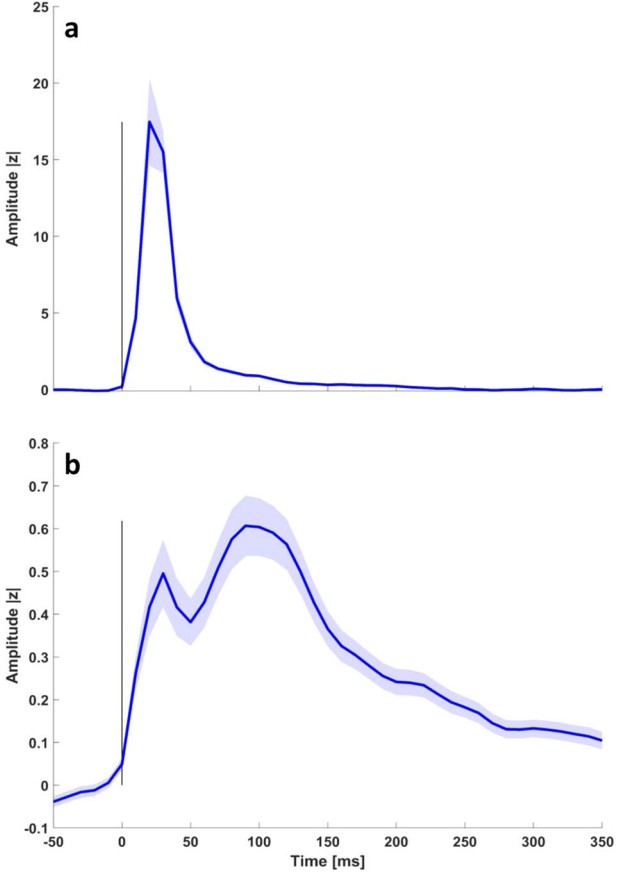

**Fig. 3 Gamma band time-course following contralateral median nerve stimulation. a** shows the gamma band temporal course of leads responsive to contralateral median nerve stimulation by leads sampling SI (12 out of 12) with error bar corresponding to standard error. **b** shows the response to contralateral median nerve stimulation of both left and right SII (±SE). For visualization purposes, data are Savitzky–Golay filtered. The average has been computed taking into account all leads (23 left, 21 right) with a statistically significant response to the stimulation (see Methods).

tie to each other in the parietal operculum. Furthermore, neuroimaging studies also demonstrated that human SII is more activated during active finger movements than during passive ones[54]. Moreover, the activation of SII is particularly strong during hand-manipulation tasks in which complex object manipulation was contrasted against a sphere manipulation[52].

Haptic function requires the presence, in addition to tactile and proprioceptive components, also of a frequently neglected motor one. The presence of this last component in SII is supported by a recent single neuron study in the monkey. Indeed, Ishida and coworkers[29] reported that some SII neurons activate during active manipulation also when tactile and proprioceptive fields are absent. The presence of neurons activated by self-movements has been reported by Hihara et al.[28], who interestingly also identified in this class of neurons most of the visual responses of SII to action observation.

Despite SI and SII exhibit a similar gamma-band temporal course during action execution, they likely underlie different functional roles, with SII activation encoding haptic functions[14,55]. A functional segregation between SI and SII is confirmed by the different reactivity of these regions to basic somatosensory stimulation. The median nerve stimulation determines a short-lasting, phasic response in the contralateral SI, whereas the same stimulation elicits a prolonged, tonic activity in both contralateral and ipsilateral SII.

This evidence is also in line with previous studies from our group[13,21,56], confirming a segregation between SI and SII within somatosensory processing.

During action observation, SII is active, while SI does not show any significant increase in gamma power respect to the baseline period, thus suggesting a lack of activation in this area[57]. The possible activity of SI during action observation is a debated issue in social neuroscience[32,58,59]. In our study, intracranial recordings were performed from areas 3a, 3b, and 1, sparing area 2. This is in line with the distinct functional roles of these sub-regions within SI complex, with visual responses limited to area 2, as indeed suggested by Keysers et al.[60].

The reactivity of SII during action observation represents the most relevant result of our study. While several studies have already reported the activation of SII during action observation[28,38,61] all these studies fail to address the role of SII region in complex actions composed by different motor acts (i.e., motor planning, reaching and manipulation). Here, we show for the first time that such responses show an identical temporal profile.

One hypothesis to explain this congruence might be that our data simply reflect a visual processing of the action, which occurs both when the action is observed and when it is performed by the subject. However, no response was found in SII following presentation of elementary visual stimuli, even when they contained visual motion, thus excluding the involvement of SII in basic visual processing. Furthermore, a recent study on monkeys identified a common coding for grasping execution and observation in SII, maintained also when the primates performed actions in the dark[61]. In conclusion, the parallelism of the SII time-courses during action observation and execution favors the hypothesis that SII plays a role in action mirroring.

This point is even stronger considering that the responsiveness to action observation in SII was found in a set of leads entirely recruited during action execution, thus witnessing a mirror-like activity in SII matching action execution and observation. The involvement of SII in action mirroring has been previously suggested by several studies[31,32,38]. One interpretation suggests that the sight of humans being touched triggers a somatosensory representation in the observer, informing her about the quality of touch she would experience if touched in a similar way. However, the notion that (a) SII hosts motor neurons in addition to neurons with tactile and proprioceptive fields[29], and (b) that observed actions are arranged in the parietal lobe (to which SII belongs in cytoarchitectonical terms) in functional clusters according to the type of observed action[31,62] allows an alternative interpretation.

Of particular interest in this context is the fMRI study by Ferri et al.[31], which demonstrated that SII activity during action observation is selective for manipulative actions, and virtually absent for observed touch, supporting the notion that this area encodes a representation of actions requiring haptic control. Our findings support this view, showing in addition that such tuning is shared with action execution. Indeed, we reported a reliable response shared by execution and observation during the object manipulation, i.e., the action phase with the strongest haptic component.

The notion that SII encodes more than just somatosensory information during executed actions is further supported by a recent study by Limanowsky and coworkers[63]. They reported that tactile inputs activate SII more strongly when in concomitance with active movements, proposing for SII a role in instantiating long-lasting sensorimotor responses to be further used by higher-order motor regions for motor adjustments.

The data presented in this study demonstrate that, besides tactile processing, SII activates during the observation of grasping and manipulation actions with a temporal profile synchronous with that shown during action execution.

These results indicate that a mirror-like mechanism is present also in SII, sustained by a neural circuit able to operate simultaneously and independently from the somatosensory input SI. The peculiarity of the SII mirror mechanism lies in its specificity for motor acts requiring haptic exploration, reinforcing the view of SII as an area functionally related to the properties of IPL.

## Methods

**Subjects**. Stereo-EEG (sEEG) data were collected from 18 right-handed patients (12 males, 6 females) suffering from drug-resistant focal epilepsy (age $36 \pm 7$)

Patients were stereotactically implanted with intracerebral electrodes as part of their presurgical evaluation, at the Centro per la chirurgia dell'Epilessia "Claudio Munari", (Ospedale Ca' Granda-Niguarda, Milan, Italy). Implantation sites were selected on clinical grounds, using seizure semiology, scalp-EEG, and neuroimaging as guide. Patients were fully informed regarding the electrode implantation and sEEG recording, and informed consent was obtained. The present study received the approval of the Ethics Committee of Ospedale Ca'Granda-Niguarda (ID 939-2.12.2013). Intracerebral recordings were performed according to sEEG methodology to define the cerebral structures involved in the onset and propagation of seizure activity. Neurological examination was unremarkable for all patients and, in particular, no patient presented any motor or sensory deficit.

**Electrode implantation**. Six patients were implanted in the left hemisphere, eight patients in the right one and four were implanted bilaterally, resulting in a total of 22 hemispheres explored. The implantation lateralization was chosen according to previous clinical investigations and ictal semiology.

A number of depth electrodes (range: 9–19; average: 14) were implanted in different regions of the hemisphere using stereotactic coordinates. Each cylindrical electrode had a diameter of 0.8 mm and consisted of eight to eighteen 2-mm-long contacts (leads), spaced 1.5 mm apart (DIXI Medical, Besancon, France). Immediately after the implantation, cone-beam computed tomography (CBCT) was obtained with the O-arm scanner (Medtronic) and registered to pre-implantation three-dimensional (3D) T1-weighted MR images. Subsequently, multimodal scenes were built with the 3D Slicer software package[64], and the exact position of each lead was determined, at the single patient level, looking at multiplanar reconstructions[65].

**Anatomical reconstruction of electrodes**. The aim of the reconstruction was to localize the recording leads in the individual cortical surfaces and, via a two-dimensional (2D) co-registration, to merge leads from all patients onto a common template to identify the ones exploring regions of interest. The procedure adopted in this study is the same as presented in[56].

**Experimental design**. Patients performed two experimental sessions: in the first they were required to perform a reach-grasp task and an ecological manipulation on different objects set in a workbench (i.e., tighten a screw, beat a nail or screw a bolt with the hand). In the second session, patients were asked to carefully observe the same task performed by an experimenter. Both patients and the experimenter performed all the experimental session with the right hand.

Session were composed by 60 trials (20 for each object, randomly sorted), each comprising three different phases whose onset and offset were signaled by digital events. While sitting in front of the workbench, the subject had first to press a button box with the right hand (four fingers, except the little finger posed on the button box) as initial position.

"Movement preparation" phase (duration 2 s): The initial position triggers the instruction about which object the patient/experimenter will have to manipulate. This information is administered by turning on a light-emitting diode (LED) under the object to-be-manipulated, remaining turned on for 2 s (see Fig. 4a).

"Reaching" phase (variable duration): as far as the LED turns off, the patient is free to start the action. The beginning of the reaching phase is identified by the button box signaling when all the buttons are unpressed, while the end is identified by a photocell on the top of the workbench (see Fig. 4) estimating the onset of the hand/object interaction. The duration of this phase varied within and across patients;

"Manipulation" phase (duration 2 s): In this phase the patient/experimenter is required to manipulate the object (beating the nail, screwing the bolt, and tighten the screw). The end of the manipulation was signaled by an acoustical tone, delivered 2 s after manipulation onset, after which the agent has to return in the starting position.

During the whole experiment, both the patient and the experimenter were required to minimize their postural adjustments.

**Clinical and neurophysiological tests**. The day after the implantation, patients were admitted to the neurology ward, to undergo clinical and neuropsychological tests to functionally characterize the recording leads.

Median nerve stimulation: the median nerve opposite to the recorded hemisphere was stimulated at the wrist, using 100 constant-current pulses (0.2-ms duration) at 1 Hz while the patient lied in bed with eyes closed. The intensity and exact site of stimulation were varied until an observable thumb twitch was obtained. The motor threshold in our sample ranged from 3.2 to 5.8 mA. The stimulation intensity was set at 10% above the motor threshold.

Acoustic stimulation: patients wearing earphones were required to listen to 100 click acoustical stimulation (contralateral to the implanted hemisphere) of, respectively, 40 and 85 dB SPL (Sound Pressure Level).

Visual stimulation: patients wearing goggles received 100 bilateral visual stimulations (i.e., flash) at a rate of 1 Hz.

Optokinetic stimulation: patients were required to sit in front of a computer screen and watch a sequence composed by six images representing concentric curves enlarging at each image and thus indicating an anterograde progression in space. The duration of the whole sequence was set to 320 seconds; the number of trials was 90.

All the stimulations were delivered by means of Nihon-Khoden Neuropack M1 stimulator, allowing to search for threshold values and control stimulation intensity and/or frequency.

**SEEG data recording and processing**

*Recordings*. For each patient, the initial recording procedure included the selection of an intracranial reference, which was chosen by using both anatomical and functional criteria. The reference was computed as the average of two adjacent leads both exploring white matter. These leads were selected time-by-time because they did not present any response to standard clinical stimulations, including

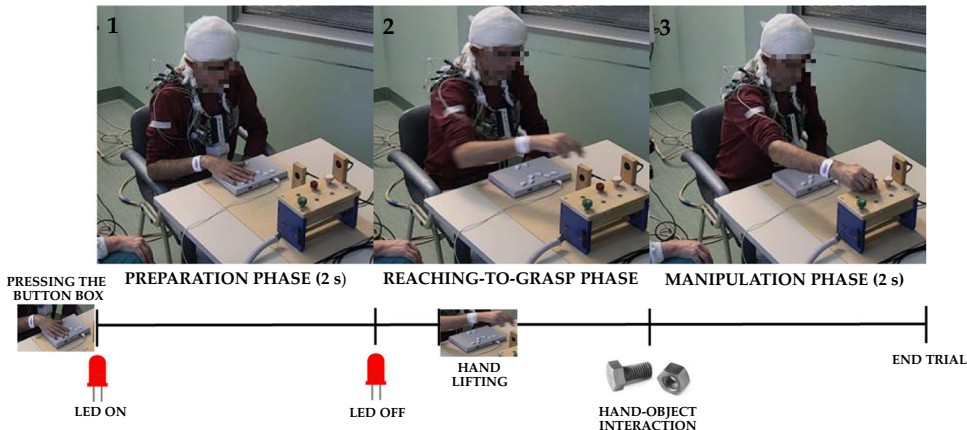

**PRESSING THE BUTTON BOX** · **PREPARATION PHASE (2 s)** · **REACHING-TO-GRASP PHASE** · **MANIPULATION PHASE (2 s)**

LED ON · LED OFF · HAND LIFTING · HAND-OBJECT INTERACTION · END TRIAL

**Fig. 4 The experimental paradigm.** The figure depicts the experimental procedure. In the preparation phase (1), the subject is required to keep a button box pressed with the right hand and to fixate a LED indicating the object to be manipulated (duration 2 s). When the led turns off (2) the subject may lift the hand (self-paced) from to the button box to reach the object (reaching-to-grasp phase). This phase ends when the subject's hand crosses a photocell placed just above the objects (hand-object interaction). Finally, in the manipulation phase the subject is required to manipulate the object continuously for two seconds until the end of the trial.

somatosensory (median, tibial, and trigeminal nerves), visual (flash), and acoustical (click) stimulations. Nor did the leads' electrical stimulation evoke any sensory and/or motor behavior. The sEEG trace was recorded with a Neurofax EEG-1100 (Nihon Kohden System) at 1-kHz sampling rate. Clinicians visually inspected recordings to verify for ictal epileptic discharges (IEDs) during the stimulation protocol. No patients presented IEDs during the recordings of the experiment.

*Data processing.* The data from all leads in the gray matter were decomposed into time–frequency plots using complex Morlet's wavelet decomposition. To avoid power-line contamination, power in the gamma frequency band was extracted from 55 to 145 Hz for each of three phases (preparation, reaching, and manipulation) in execution and observation conditions. Gamma band power was also computed for baseline condition ranging from 350 ms to 50 ms before the led lighting. In median nerve, acoustical and visuo-motor stimulation, the selected window for gamma-band power computation ranged from 100 ms before to 500 ms after the stimulus delivery. Finally, gamma power was subdivided into non-overlapping 10-ms bins and estimated for ten adjacent 10-Hz frequency bands[42,66].

To compare the gamma-band power dynamics during the reaching phase within and across subjects, the estimation in each frequency band has been linearly interpolated in a fixed number of points ($n = 155$).

**Statistics and reproducibility**. To identify the responsive leads, the gamma band power in each post-stimulus bin was compared with baseline using a $t$-test. Significance was corrected for 50 comparisons ($p = 0.001$), and to decrease the false-positive ratio, only leads with significant gamma increases in at least three time bins were designated as responsive. Note that significance for the reaching phase was computed limiting the comparison at the minimum common duration across trials for each patient, independently for execution and observation conditions. To normalize data across patients and leads, power in post-stimulus bins was transformed into $z$-scores relative to the baseline interval. Then, a one-sample $t$-test was computed to determine time bin significance across population of responsive leads. Furthermore, a two-sample $t$-test compared (a) the gamma band activity between contralateral SI and SII for both execution and observation condition; (b) the gamma band activity between left SI and SII in execution condition (for both raw data and after normalization per leads in the range 0–1; (c) the gamma band modulation between contralateral and ipsilateral SII separately for execution and observation condition. For each phase, significance was Bonferroni corrected. Finally, we estimated the delay between responses following action execution and observation as the first significant bin of activation after hand lifting; significance against the zero is then estimated by a one-sample $t$-test.

Both data processing and statistical analysis were performed with Matlab R2011b. Mapping of spatial sampling and responsiveness maps were computed according to the procedures detailed in[66] and visualized on a flat map with Caret software[67]. The correspondences between the areas depicted on a flat map with an inflated model of brain are reported in Supplementary Fig. 1.

**Reporting summary**. Further information on research design is available in the Nature Research Reporting Summary linked to this article.

## Data availability
The data that support the findings come from a clinical population, who voluntarily participated in this study. All data are available from the corresponding author upon reasonable request.

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

## Acknowledgements
M.D.V. was supported by a grant POR-FSE 2014-2020 funded by Regione Emilia Romagna (Italy) and by European Union Horizon 2020 Framework Programme through Grant Agreement No.785907 (Human Brain Project, SGA2) to P.A.

## Author contributions
F.C., P.A., G.R. conceived and designed the experiment. I.S., V.P., G.L.R. performed the experiment. M.D.V. performed the analysis. M.D.V., F.C., G.R., P.A. wrote the manuscript. All authors reviewed and edited the final manuscript.

## Competing interests
The authors declare no competing interests.
