## [Peer Review File · Communications Biology]

Reviewers' comments:

Reviewer #1 (Remarks to the Author):

This work uses a highly specialized methodology to record activity in the secondary somatosensory cortex in the human brain during both action execution and action observation. The findings presented here suggest powerful and impactful data which can be used as evidence to motivate future work in the action-observation and mirror neuron system fields. There are some topics which could be engaged in more deeply to improve the quality of this particular manuscript.

Line numbers would be helpful for reviewing purposes, in addition to page numbers which are also not present. This has limited my ability to provide a detailed review.

Abstract: wording and phrasing are off: "requiring a revision of the notion that human SII is purely unimodal sensorimotor."

3rd page of document: The authors state that it is impossible to investigate active movement during imaging, however this type of work has been done many times with respect to mirror neuron and action observation processes. There are still clear reasons why intracortical EEG would be beneficial, but the current manuscript falls short of accurately describing those reasons.

There is no background on why gamma range EEG is used in this study. The reader would benefit from knowing something about this methodology.

Stimuli are not described clearly. What were the objects used by participants? The description of the task is unclear.

The Results section does not quantitatively present the significance of the values shown. Were statistical tests performed? How likely is it that these findings could occur by chance? Some results are described narratively, but the reader is left to wonder if the patterns being described are the result of random variation (e.g., "generally lower amplitude of the activations") or whether they are significant findings.

Other research has suggested that SII plays an especially important role in action mirroring. It would be helpful if the reviewers explicitly state what new evidence they present here, and how it helps to strengthen this existing idea.

Unclear use of the term imaging: do they mean fMRI, or all forms of imaging?

Subject-verb agreement throughout could use revision.

Figure 2 lacks any letters though the letters A and B are referenced in the caption.

In Figure 3, there are two panels but they are not labeled.

Regarding the acoustical stimulation, the phrase "of respectively 40 and 85 dB" does not refer to anything, so it is not clear what the 40 and 85dBs are describing.

Table 1: Is this data for SI or SII?

Reviewer #2 (Remarks to the Author):

The authors describe the results of one experiment in which stereo-EEG was recorded in human participants performing and observing actions composed by a planning, reaching and manipulation phase, with the aim of studying SI and SII role in actions execution and observation.

While the main idea that SI/SII activity may have a role during observation of hand actions is not so novel in light of previous studies testing this hypothesis (Avikainen et al, 2001; Gazzola and Keysers 2009), the possibility to study the temporal dynamics of SI/SII activation during action observation and execution is relevant. I think that the authors should clarify some theoretical and methodological aspects. Here are my comments:

-I think that the hypotheses should be better clarified in the Introduction: were the authors expecting differences in gamma band activity during the planning, reaching and manipulation phase during execution compared to observation? Were they expecting differences in gamma band activity for SI and SII? Please explain;

-The focus on gamma band activity should be justified in the Introduction;

-As far as the SI activity is concerned, I think that the authors should treat the analysis of gamma activity of SI and the relative results as preliminary, since only 4 patients were involved;

-The authors write that SII was explored in 16 patients and SI in 4. Were the patients whose implantation cover SI also having implantation covering SII? Please explain;

-In the results section please report all the p, t, degree of freedom and effect size of the t test;

-In the Introduction the authors state that "A strong limitation of imaging studies concerns the impossibility to investigate active complex movements, hence preventing any comparison between sensorimotor responses..." However other studies did so (for example Avikainen et al.. 2001), thus the authors should better explain the limitation they are introducing and how their study overcomes this limitation.

Reviewer #4 (Remarks to the Author):

This manuscript builds on the wealth of data surrounding the secondary somatosensory cortex (SII), and makes use of stereotactic EEG to directly compare SII activation during both observation and execution of actions. Clinical patients were tested using intracranial electrodes while performing, and subsequently observing, a series of grasping and manipulating actions. Time-frequency transformations were performed to allow the assessment of gamma power across electrode sites and over time. In addition, simple stimulation tests were conducted to observe SI and SII responsiveness to different low-level stimuli, including touch, vision and audition. Overall, this study demonstrates that while SI is active only during action execution, SII activity is remarkably similar (both in temporal pattern and in electrode sites showing activation) during both execution and observation of the same actions. In addition, the authors ruled out any low-level visual interpretation of SII activity, and

demonstrated functionally distinct processing of the same stimulation by SI and SII.

It was a real pleasure to read this paper; coming from a background in Cognitive Neuroscience and Social Neuroscience and working mostly at a systems level rather than an anatomical level, the topic of this paper was initially daunting to me. However, in my opinion the authors succeeded in presenting both the background literature and their own research in a detailed yet digestible manner.

Below are some points that I believe the authors should address or consider:

1) Generally, there appeared to be a lack of statistical comparisons between conditions. As far as I could tell, the only statistical tests performed were t-tests comparing gamma power during baseline vs. action, to determine if a given lead was significantly responsive. However, at several points the authors draw comparisons between conditions. For example, in lines 323-328, the authors say that "The activity pattern during action observation was highly similar [...] but with a generally lower amplitude of the activations and a [...] delay of 50 ms relative to action execution". These differences are visually clear when consulting Figure 2, but it is impossible to know whether there is a statistically significant difference in either gamma power amplitude or temporal onset across observation and execution conditions. Ideally, the authors should run a statistical test to determine this.

2) Following on from this, I would also recommend including a statistical comparison of SI and SII activity; this may be useful in demonstrating that there is no difference in gamma power between these areas during action execution, but a significant difference during action observation. This is not necessarily required to enforce your conclusions as your claims ultimately refer to the presence or absence of responsiveness rather than the power amplitude, but I believe it would add to your results.

3) There is a clear sample size difference in SI and SII patients (and leads). While a balanced sample size is difficult given the clinical and invasive nature of the data collection, the authors should at least acknowledge any potential implications of this sample size difference, especially if they intend on including a statistical comparison across SI and SII electrodes. In addition, the total sample size should be justified in some way, whether through a power analysis or through comparison of previous sEEG studies. Currently, the authors claim that "Taken together, these numbers provide a reliable sampling of the regions of interest" (lines 288-289), but there is no justification for this claim.

4) Are there any implications of lacking any ipsilateral SI electrode sites? I am unfamiliar with the literature on the architecture of SI and SII, but from Figure 2 it appears that there may be differences in SII activity across hemispheres; in particular, there is a generally lower amplitude in ipsilateral SII, and also a smaller amplitude difference across action conditions for ipsilateral, relative to contralateral, SII. Given this, is it possible that the relationship between observation and execution may differ across hemispheres for SI? Perhaps the authors know of literature demonstrating that it is safe to assume that SI activity is identical across hemispheres. If so, they should report this literature; if not, they should carefully address any potential implications from lacking data on ipsilateral SI.

In addition, I have some more minor points worth considering:

5) Error bars are visible in Figures 2, 3, and 4, but these are not defined in the figure legends.

6) Did the authors consider the possibility of counter-balancing the order of the execution and observation tasks? It may be interesting to see if there are differences in SII responsiveness when performing a task that patients have just observed (i.e., observation first), compared to when observing a task that patients have just performed (i.e., execution first, as was done in this study). For example, the fact that the observed actions were recently performed by the patients themselves

may influence the self-relevance of these observed actions, which could have some high-level implications for processing in regions which might convey information to SII.

7) In Table, the legend states that "no leads were responsive only in the observation condition". This appears to contradict the table itself, as it shows that 1 ipsilateral lead was responsive in the observation only condition during the reach-to-grasp phase. Please ensure this information is reported correctly.

8) The image in Figure 1 appears to be somewhat degraded and unclear, as if it has been photocopied. Is this the intended image?

Finally, a note about your interpretation of the SII responsiveness to median nerve stimulation:

9) I am very satisfied with how you interpret the large differences in SI vs. SII activation in response to this nerve stimulation, and its integration into your conclusions regarding the different functional properties of SI and SII during action execution. However, the authors may also like to consider higher-level concepts such as the Sense of Agency (i.e., the sense that "I am performing this action"). Assuming that SII is implicated in higher-level control of actions (as you have stated), it may be interesting to compare SII activation during actions that are specifically executed by the patient, and actions that are externally-stimulated. This could either be nerve stimulation, or simply an experimenter physically moving the arm and hand of the patient to grasp an object. Would SI/SII activation be similar during these two types of 'executed' actions, or would an externally-controlled action produce neural responses similar to observing another's action?

Aside from all of these points, I was very impressed with the manuscript. It is well placed within the literature and fulfills the need to elucidate the function and anatomy of SII, and its results have a clear impact on future models of the lateral grasping network and of action-mirroring.

*To submit your data to Dryad, please use the link below, which will take you directly to your entry in the Dryad repository. Using this link will ensure that reviewers will have access to your data during the review process:

<http://datadryad.org/submit?journalID=COMMSBIO&manu=COMMSBIO-19-0880-T>

Once your data package is deposited, a unique and stable identifier (DOI) will be sent to you and to the journal for inclusion in the published article. For questions or feedback on the Dryad data submission process, please email help@datadryad.org.

Reviewer #1 (Remarks to the Author):

This work uses a highly specialized methodology to record activity in the secondary somatosensory cortex in the human brain during both action execution and action observation. The findings presented here suggest powerful and impactful data which can be used as evidence to motivate future work in the action-observation and mirror neuron system fields. There are some topics which could be engaged in more deeply to improve the quality of this particular manuscript.

Line numbers would be helpful for reviewing purposes, in addition to page numbers which are also not present. This has limited my ability to provide a detailed review.

Line numbers as well as page numbers have been added to the manuscript.

Abstract: wording and phrasing are off: “requiring a revision of the notion that human SII is purely unimodal sensorimotor.”

Thanks for the suggestion. The word unimodal has been now removed from the whole text. More importantly, when necessary, we detailed the sensorimotor functions explicitly along the manuscript (i.e. tactile, proprioceptive).

3rd page of document: The authors state that it is impossible to investigate active movement - during imaging, however this type of work has been done many times with respect to mirror neuron and action observation processes. There are still clear reasons why intracortical EEG would be beneficial, but the current manuscript falls short of accurately describing those reasons. *The movements used in our experimental paradigm are not only active, but also performed in ecological settings. Subjects were free to explore the space and observe the experimenter in a completely natural way. Most of the scientific literature investigated human mirror mechanism process by using fMRI, in which subjects are required to lay in in the scanner, obtaining visual feedback by a mirror, placed above subjects' head. Notwithstanding, although electrophysiological techniques (e.g scalp EEG) might overcome this limitation, they are affected by a poor source localization which prevents from studying the responsiveness of specific areas of the brain, in a consistent way across subjects. Intracortical EEG combines, instead, a high spatial and temporal resolution with the possibility to study the full and natural dynamics of an action. The introduction of the manuscript has been thus modified to fulfill this point (lines: 119-134)*

There is no background on why gamma range EEG is used in this study. The reader would benefit from knowing something about this methodology.

Thanks for this point. In lines 134-136, we specified the reasons for concentrating on gamma-band modulation. In fact, it is agreed that gamma rhythm serves as indicator of neuronal activation reflecting spiking activity (Manning et al. 2009, Vangeneugden et al. 2008) and being highly functionally and spatially specific (Lachaux et al. 2012, Vidal et al. 2010).

Stimuli are not described clearly. What were the objects used by participants? The description of the task is unclear.

Participants, as well as the experimenter, performed the experimental task only by using their right hand. Both of them performed the hand-object interaction in a completely ecological way suggested by the object (i.e. tighten the screw, beat the nail and screw the bolt). The description of the experimental procedure has been modified and it is now reported in more detail (lines 194-221). In addition, figure 1 has been updated, adding also a paradigm timeline detailing the phases of the experiment.

The Results section does not quantitatively present the significance of the values shown. Were statistical tests performed? How likely is it that these findings could occur by chance? Some results are described narratively, but the reader is left to wonder if the patterns being described are the result of random variation (e.g., “generally lower amplitude of the activations”) or whether they are significant findings.

We apologize for having not provided the full statistical picture of our results.

The Results section has now been updated, adding statically relevant information. In particular:

- *Lines 377-380 - concerning contralateral SII the manuscript reported: “The activity pattern during action observation was highly similar to that observed during action execution, but with a generally lower amplitude of the activations”. We now assessed similarity between the two conditions (execution - observation) for both contralateral and ipsilateral SII by computing a linear regression fitting between the averaged z-score amplitude over time.*

We obtained the following parameters:

- *R-squared: 0.513, F-statistic vs. constant model: 637, p-value < 0.001 for contralateral SII;*
- *R-squared: 0.695, F-statistic vs. constant model: 1380, p-value < 0.001 for ipsilateral SII;*

Figure 2 has also been updated reporting the linear model as well as the confidence interval for both hemispheres (panels C and D).

- *Lines 381-382 – “with a generally lower amplitude of the activation” – Figure 3 in Supplementary Materials reports a t-test to compare execution and observation conditions for contralateral (panel A) and ipsilateral (panel B) SII. Significance is corrected with Benferroni for each phase. While left SII presents differences in amplitude (z-score) between the two conditions, significance can be considered residual for the right hemisphere.*
- *Lines 383-386 “response onset following the hand lifting (delay of 50 ms relative to action execution in terms of earliest significance for leads active both during execution and observation conditions)” Delay between execution and observation conditions has been reported for each lead in table 2, and its significance has been computed via a one-sample t-test against zero ($p < 0.001$).*

Other research has suggested that SII plays an especially important role in action mirroring. It would be helpful if the authors explicitly state what new evidence they present here, and how it helps to strengthen this existing idea.

Several studies have already reported the activation of SII during action observation (e.g. Avikainen et al. 2001, Ishida et al. 2011, Hihara et al. 2015, Fiave et al. 2018). These studies were performed by using different recording techniques, ranging from single neurons recording in the monkey to human fMRI and magnetoencephalography. Despite their fundamental role in demonstrating the role of SII in action mirroring, these studies fail to provide a comprehensive figure of the role of SII region in complex actions, including motor planning, reaching and manipulation. Our study instead investigated, in out-of-the-lab-like conditions, the contribution of SII to the full dynamics of a reaching-to-grasp and manipulative action, both during execution and observation.

Accordingly, the specific contribution of this study goes far beyond the mere demonstration of an overall responsiveness of SII to both action execution and observation – which was indeed already known thanks the studies mentioned above. Rather, it provides the first evidence of the synchrony of the SII gamma band time-course in both conditions. This finding allows for additional considerations.

While the temporal pattern shown by this area during execution might reflect an input from SI, this cannot be true during the observation, due to the lack of any activation in SI. Hence, during action observation, SII activity is sustained by a neural circuit able to operate simultaneously and independently from the somatosensory input. Secondly, the sustained activation of SII, starting before hand-object interaction and lasting for the whole manipulation period, indicates that this area might specifically mirror functions of haptic exploration.

MINOR

Unclear use of the term imaging: do they mean fMRI, or all forms of imaging?

Imaging-related terms have been disambiguated all along the text.

Subject-verb agreement throughout could use revision.

Subject-verb agreement has been checked all along the text.

Figure 2 lacks any letters though the letters A and B are referenced in the caption.

In Figure 3, there are two panels but they are not labeled.

Captions and labels have been corrected.

Regarding the acoustical stimulation, the phrase “of respectively 40 and 85 dB” does not refer to anything, so it is not clear what the 40 and 85 dBs are describing.

40 and 85 dBs SPL (Sound Pressure Level). In the related section of the Methods we now better clarify how acoustical stimulation was delivered to the patients.

Table 1: Is this data for SI or SII?

Data are related to SII; caption of the table has been made clearer.

Reviewer #2 (Remarks to the Author):

The authors describe the results of one experiment in which stereo-EEG was recorded in human participants performing and observing actions composed by a planning, reaching and manipulation phase, with the aim of studying SI and SII role in actions execution and observation.

While the main idea that SI/SII activity may have a role during observation of hand actions is not so novel in light of previous studies testing this hypothesis (Avikainen et al, 2001; Gazzola and Keysers 2009), the possibility to study the temporal dynamics of SI/SII activation during action observation and execution is relevant. I think that the authors should clarify some theoretical and methodological aspects. Here are my comments:

- I think that the hypotheses should be better clarified in the Introduction: were the authors expecting differences in gamma band activity during the planning, reaching and manipulation phase during execution compared to observation? Were they expecting differences in gamma band activity for SI and SII? Please explain;

- *In Lines 137-148 of the Introduction, we now elucidate both the aims and the hypothesis of our study. As far as we know, although it is well-known that both SI and SII play a role in action execution, their relationship have not been fully addressed in terms of temporal dynamics during actions (i.e. during motor planning, reaching-to-grasp and manipulation). This point is even more true if considering action observation: several studies (e.g. Avikainen et al. 2001, Ishida et al. 2011, Hihara et al. 2015, Fiave et al. 2018) have already reported the activation of SII, without however clarifying the functional role subtending this responsiveness. For these reasons, we are interested in addressing the relative temporal dynamics of responsiveness of SI and SII during action execution and action observation, which might provide fundamental elements to disentangle the contribution of this area in mirror mechanism.*

- **The focus on gamma band activity should be justified in the Introduction;**

Lines 134-136: We clearly stated in the Introduction the reasons subtending the adoption of gamma-band modulation as indicator of SI/SII activity. It is well-known that gamma-band activity represents a reliable indicator of neural activation: in fact, it has been reported to reflect spiking activity (Manning et al. 2009, Vangeneugden et al. 2008) and to be highly functionally and spatially specific in several studies (Lachaux et al. 2012, Vidal et al. 2010).

- **As far as the SI activity is concerned, I think that the authors should treat the analysis of gamma activity of SI and the relative results as preliminary, since only 4 patients were involved;**

Lines 436-437: In the Discussion section, we now admit that the small number of patients whose implantation covered contralateral SI represents a limitation for this study. It is worth noting, however, that all the 12 leads included in the study display a common temporal behavior in our experimental paradigm (in figure R1 - panel A: action execution, panel B: action observation) thus

suggesting that the picture stemming from this data can be considered sufficiently reliable. Concerning the behavior shown after contralateral median nerve stimulation, our data fit very well with the phasic cluster described in Avanzini et al. (2016, 2018) which is peculiar of the primary somatosensory cortex (SI)

Figure R1

- The authors write that SII was explored in 16 patients and SI in 4. Were the patients whose implantation cover SI also having implantation covering SII? Please explain.

Lines: 341-345. Methods section includes now more details regarding patient implantation: "SII (but not SI) was explored in 14 patients (5 left, 8 right and 1 bilaterally), 2 patients' implantation covered left SI (but not SII), while other 2 patients showed implantations exploring both SII (bilaterally) and left SI. As far as individual leads are concerned, 63 leads recorded activity from SII (31 left, 32 right), while 12 from left SI."

- In the results section please report all the p, t, degree of freedom and effect size of the t test;

We thank the Reviewer for raising this point. For the convenience of the reviewer, in the table below we reported the values of t, degree of freedom and effect size (Cohen's d) of the t-test. Being t-tests computed time-wise, we reported the range for each phase of responsiveness. Minimum p-values are corrected for Bonferroni.

Test	t range			minimum p-value (Bonferroni corrected)			dof	effect size (Cohen's d)		
	PLAN	RTG	MANIP	PLAN	RTG	MANIP		PLAN	RTG	MANIP
Contralateral SII, execution (against the baseline)	[-2.57, 3.13]	[0.88, 15.62]	[5.67, 16.12]	1.02	< 0.001	< 0.001	21	[-0.55, 0.67]	[0.19, 3.32]	[1.21, 3.44]
Ipsilateral SII, execution (against the baseline)	[-3.91, 3.32]	[-1.27, 11.39]	[3.74, 12.35]	0.20	< 0.001	< 0.001	18	[-0.9, 0.76]	[-0.29, 2.61]	[0.86, 2.83]
Contralateral SII, observation (against the baseline)	[-1.65, 7.75]	[0.31, 10.27]	[0.63, 8.37]	< 0.001	< 0.001	< 0.001	21	[-0.35, 1.65]	[0.07, 2.19]	[0.14, 1.78]
Ipsilateral SII, observation (against the baseline)	[-4.23, 5.06]	[-0.98, 12.73]	[1.84, 12.13]	0.07	< 0.001	< 0.001	18	[-0.97, 1.16]	[-0.23, 2.93]	[0.42, 2.78]
Contralateral SII (execution vs observation)	[-3.76, 2.52]	[-1.08, 8.24]	[-2.05, 8.42]	0.10	< 0.001	< 0.001	41	[-1.13, 0.69]	[-0.32, 2.49]	[0.66, 2.45]
Ipsilateral SII (execution vs observation)	[-4.06, -3.02]	[-2.91, 4.07]	[-2.66, 4.93]	0.05	< 0.001	< 0.001	37	[-1.32, 0.84]	[-0.94, 1.32]	[-0.39, 1.6]

- In the Introduction the authors state that “A strong limitation of imaging studies concerns the impossibility to investigate active complex movements, hence preventing any comparison between sensorimotor responses...” However other studies did so (for example Avikainen et al. 2001), thus the authors should better explain the limitation they are introducing and how their study overcomes this limitation.

Lines 119-134: The introduction has been modified to clearly indicate the advantages of intracortical recordings during the investigation of active movements. Although other human studies employed manipulation tasks (both in execution and observation conditions), these were unable to show the whole dynamics sustaining complex actions, focusing rather on specific actions timing or single motor acts (see Gangitano et. al 2001). Taken together, these experimental procedures allows for the implementation of a truly ecological paradigm: no videos were shown during the observation condition, and subjects were free to explore the surrounding space naturally while they get unmediated visual feedback.

Reviewer #3 (Remarks to the Author):

This paper, titled “More than just somatosensory: intracortical responses of SII to action observation” demonstrates, for the first time in stereo-EEG in humans, that SII responds similarly to SI for active reaching movements. The authors also show evidence that supports previous studies that SII responds in a similar fashion when either a subject or experimenter is performing the task which the subject is observing. The study shows some intriguing results that are of broad interest to the field, but would also benefit from several changes and additions. One of the major claims of the manuscript is that SII should no longer be considered as a unimodal somatosensory area, which, based on previous findings and the results presented here, seems warranted.

We thank the reviewer for the positive comments on the manuscript.

However, the definitions of uni- and multi-modal seem largely unclear. For example, in lines 101-105, active and movements, proprioception, and tactile stimuli are presented as separate modalities. While this could potentially be justified, S1 would also need to be considered multimodal as it contains cutaneous (areas 3b, 1, 2) and proprioceptive (areas 3a, 2) receptive fields. Because S1 projects to SII, SII is expected to be multimodal by nature. Therefore, the manuscript would benefit from a clear definition of “modes” of sensory inputs, as this is important to one of the major claims of the paper.

We agree that the terms “unimodal” and “multimodal” can raise misinterpretations. For this reason we removed them from the text, using the specific term (i.e. tactile or proprioceptive) according to the case.

It is also somewhat difficult to pick out the question being asked by the authors or the unique contribution of his paper to the field that has not been shown previously. From early on in the introduction (lines 104-105), the authors mention electrophysiology studies in monkeys that show SII neurons respond to observation of reaching and grasping, but later (122-123) claim the objective of this paper is to elucidate whether SII is recruited during action observation, which is apparently already known. Therefore, it would be beneficial to more clearly define the novel contribution of this paper to the field throughout the manuscript.

Lines 119-134, Lines 534-539: It is true that several studies have already reported the activation of SII during action observation (e.g. Avikainen et al. 2001, Ishida et al. 2011, Hihara et al. 2015, Fiave et al. 2018). However, all these studies fail to address the role of SII region in complex actions composed by different motor acts (i.e. motor planning, reaching and manipulation). Our study investigated, in ecological conditions, the contribution of SII to the full dynamics of an action, both during execution and observation. Apart from the overall responsiveness of SII to the execution and observation of the same action, we show for the first time the identical temporal profile of the SII gamma band time-course in both conditions. Furthermore, the absence of activation of SI in the

observation condition indicates that, SII responses are sustained by a neural circuit able to operate simultaneously and independently from the somatosensory input. Finally, we can speculate that the sustained activation of SII during the whole manipulation period reflects the involvement of SII in specifically mirroring haptic functions.

The activity analyzed in this paper is the gamma-band limited activity recorded from the stereo-EEG electrodes, but the implications of this choice are not considered. Because of this, many of the conclusions of the study are too strong (e.g., line 423: “SI is completely unresponsive”). In fact, you only analyzed one band of frequencies out of many that may carry neural responses to the stimuli in this study. As such, a discussion of the potential effects of this decision on your outcomes is necessary.

Thank you for your point. However in the introduction section we clearly state that we focus on studying gamma band modulation since it has been considered to be a reliable index of neuronal spiking (Manning et al. 2009, Vangeneugden et al. 2008). Furthermore, it has been reported that gamma band response shows strong category-specificity to stimuli, also at a single trial level (Vidal et al. 2010). Taking together these evidence, we concluded that gamma band responses would provide a reliable picture of the role of SII in executed and observed complex actions.

Minor comments:

Introduction

Lines 101-105: How are tactile inputs, active hand movements, and proprioception distinct in this context?

In these lines we referred to the studies by Fitzgerald et al. (2004), Hishida et al. (2011) and Hihara et al. (2015); in the context of our research, instead, we aimed at addressing the behavior of the SII in complex actions embedding a motor planning (first phase of paradigm), proprioception (e.g. during reaching-to-grasp) and haptic components, this latter representing the main feature of the manipulative phase.

Lines 109-110: Does “embedding SII also in visuomotor transformation” mean that SII could have a functional role in visuomotor transformations?

Yes, this is what we meant with that sentence. We now clarified in the text that SII has a functional role in visuomotor transformations.

Methods

Line 178, 181, others: Presumably, this should be LED. Corrected

Results

Line 313: Where is this referring to? If it is simply the following line, it is not necessary.

Corrected

Line 331-332: How was this tested?

*Thank you for this point. Figure 5 of the Supplementary Materials now statistically compares (*t*-test, Bonferroni corrected for number of comparisons for each phase of the paradigm), the responsiveness between contralateral and ipsilateral hemisphere for execution (panel A) and observation conditions (panel B). Statistically significant differences are residual (see figure R2, plotted below).*

Figure R2

Figure 1: It is very difficult to see the task in these images due to the high contrast filtering. Presumably, this was done to avoid identifying features of the subject, but perhaps this could be done with a blur effect instead of contrast manipulation.

We apologize for the lack of clarity of the previous version of Figure 1, it has now been updated. We decided to leave it in colors instead of blurring panels by contrast enhancement. Furthermore, to better clarify the trial procedures and events, we added a timeline detailing each phase of the experimental paradigm.

Figure 2: There are no labels for panels A or B. Additionally, it is difficult to tell from the figure caption what exactly is being plotted. Presumably, z-scored gamma-band electrical activity from responsive electrodes? It would be helpful to state this directly

in the figure, caption, or both. Figure 3: Similarly to Figure 2, it would be helpful to state what is being plotted.

We have now updated the captions of Figure 2 and 3.

Figure 4B: It is unclear why these data were split into left and right SII, instead of ipsi- or contralateral to the median nerve stimulation, or even keeping them together.

Thank you for this point. Following the suggestion of the Reviewer, we averaged both left and right gamma-band time courses of responsive leads following contralateral median nerve stimulation. However, we decided to keep separate left and right leads in Figure 2 of the Supplementary Material. The aim was to underline that the leads discussed in the results show limited responses to other sensory inputs (i.e. acoustic, visual and optokinetic) if compared to tactile stimulation.

Figure 5: The purpose of this figure is not totally clear to me for several reasons:

- In general, this figure is not helpful for understanding this study. Information in the caption seems to be at odds with the image, and the information in the image could probably be more compactly described in a table. If this figure is to remain, several things should change.

- What of this is hypothesized and what is known from previous research?

- The caption states "Notably, the identified network are [sic] different: SI complex represents the main source of tactile and proprioceptive information for SII while a likely input during observation might be PFG..." The identified network is different from what? If PFG is the likely input to SII during action observation, why is it red in the figure and not green? If SII receives inputs from MT by way of PFG, why is MT included in the figure?

- This would benefit from more of a flow-chart organization than how it is organized currently. Without arrows, it is difficult to see the hypothesized (?) pathways being described in the caption.

Thanks for raising this point. Following the reviewer's suggestion, we opted for removing the figure from the text, as it would add little information to the current version of the manuscript.

Table 1: I can infer what the abbreviations are, but they should be defined in the caption. In the current version of Table 1, all abbreviations have been spelled out in the caption.

Reviewer #4 (Remarks to the Author):

This manuscript builds on the wealth of data surrounding the secondary somatosensory cortex (SII), and makes use of stereotactic EEG to directly compare SII activation during both observation and execution of actions. Clinical patients were tested using intracranial electrodes while performing, and subsequently observing, a series of grasping and manipulating actions. Time-frequency transformations were performed to allow the assessment of gamma power across electrode sites and over time. In addition, simple stimulation tests were conducted to observe SI and SII responsiveness to different low-level stimuli, including touch, vision and audition. Overall, this study demonstrates that while SI is active only during action execution, SII activity is remarkably similar (both in temporal pattern and in electrode sites showing activation) during both execution and observation of the same actions. In addition, the authors ruled out any low-level visual interpretation of SII activity, and demonstrated functionally distinct processing of the same stimulation by SI and SII.

It was a real pleasure to read this paper; coming from a background in Cognitive Neuroscience and Social Neuroscience and working mostly at a systems level rather than an anatomical level, the topic of this paper was initially daunting to me. However, in my opinion the authors succeeded in presenting both the background literature and their own research in a detailed yet digestible manner.

Below are some points that I believe the authors should address or consider:

1) Generally, there appeared to be a lack of statistical comparisons between conditions. As far as I could tell, the only statistical tests performed were t-tests comparing gamma power during baseline vs. action, to determine if a given lead was significantly responsive. However, at several points the authors draw comparisons between conditions. For example, in lines 323-328, the authors say that "The activity pattern during action observation was highly similar [...] but with a generally lower amplitude of the activations and a [...] delay of 50 ms relative to action execution". These differences are visually clear when consulting Figure 2, but it is impossible to know whether there is a statistically significant difference in either gamma power amplitude or temporal onset across observation and execution conditions. Ideally, the authors

should run a statistical test to determine this.

Thank you for this point. Please, see figure 3 of the Supplementary Materials (also below). In this figure we assessed with a *t*-test the statistical differences between execution and observation for both contralateral (panel A) and ipsilateral (panel B) SII (Bonferroni corrected for comparisons in each phase). While right SII does not virtually present any difference in amplitude (z-score), left SII presents differences for both reach-to-grasp and manipulation phase. Concerning the delay, we now report its values between execution and observation condition for each lead significant in both conditions in table 2. A *t*-test was also conducted to assess its significance against the zero ($p < 0.001$)

Figure R3.

2) Following on from this, I would also recommend including a statistical comparison of SI and SII activity; this may be useful in demonstrating that there is no difference in gamma power between these areas during action execution, but a significant difference during action observation. This is not necessarily required to enforce your conclusions as your claims ultimately refer to the presence or absence of responsiveness rather than the power amplitude, but I believe it would add to your results.

Lines 446-447: Thank you for the point. We stated that left SI and SII share a similar temporal pattern during action execution. To assess this similarity, we performed a *t*-test for raw gamma-band time-course (panel A) and for normalized traces (0 – 1, panel B). Significance threshold is corrected for Bonferroni across comparisons. Figure is shown below and included in Supplementary Material as Figure 4. As it is possible to note in panel B, any significant difference disappears when

considering normal rather than for diff

amplitude modulation

Figure R4.

3) There is a clear sample size difference in SI and SII patients (and leads). While a balanced sample size is difficult given the clinical and invasive nature of the data collection, the authors should at least acknowledge any potential implications of this sample size difference, especially if they intend on including a statistical comparison across SI and SII electrodes. In addition, the total sample size should be justified in some way, whether through a power analysis or through comparison of previous sEEG studies. Currently, the authors claim that "Taken together, these numbers provide a reliable sampling of the regions of interest" (lines 288-289), but there is no justification for this claim.

In lines 436-437 we now acknowledge that the sampling obtained from SI and SII is not balanced. It is worth to underline, however, that although the rather limited amount of leads exploring SI, their behavior is highly consistent (see figure below), thus providing a quite reliable picture of SI reactivity. Moreover, an indirect cross-validation about the physiological behavior of these few SI leads is provided by their response to contralateral median nerve stimulation. Indeed, their response fits well with the strong phasic cluster described in (Avanzini et al. 2016, 2018) which identified the peculiar behavior of primary somatosensory cortex.

Figure R1

4) Are there any implications of lacking any ipsilateral SI electrode sites? I am unfamiliar with the literature on the architecture of SI and SII, but from Figure 2 it appears that there may be differences in SII activity across hemispheres; in particular, there is a generally lower amplitude in ipsilateral SII, and also a smaller amplitude difference across action conditions for ipsilateral, relative to contralateral, SII. Given this, is it possible that the relationship between observation and execution may differ across hemispheres for SI? Perhaps the authors know of literature demonstrating that it is safe to assume that SI activity is identical across hemispheres. If so, they should report this literature; if not, they should carefully address any potential implications from lacking data on ipsilateral SI.

As largely reported in scientific literature, contrary to SI, SII has bilateral receptive fields and receives somatosensory input mainly from VPL nuclei of the thalamus and from contralateral SI (Simoes et al. 2002, Dijkerman et De Haan et al. 2007, Del Vecchio et al. 2018). For this reason, our interest has been focused on left SI, with the aim to investigate possible relationships with the

activation of SII, both in execution and in the observation condition. We now clearly state in lines 114-115 of the Introduction the reasons of lacking right SI in our analysis.

In addition, I have some more minor points worth considering:

5) Error bars are visible in Figures 2, 3, and 4, but these are not defined in the figure legends.

Figures and captions have been corrected.

6) Did the authors consider the possibility of counter-balancing the order of the execution and observation tasks? It may be interesting to see if there are differences in SII responsiveness when performing a task that patients have just observed (i.e., observation first), compared to when observing a task that patients have just performed (i.e., execution first, as was done in this study). For example, the fact that the observed actions were recently performed by the patients themselves may influence the self-relevance of these observed actions, which could have some high-level implications for processing in regions which might convey information to SII.

All the subjects performed the experimental session in blocks, with the execution first. This choice has been made to avoid emulative processes for the patients, who, in this way, are free to elaborate their own motor strategies. Notwithstanding, we agree that SII response (above all during the manipulative phase of the paradigm) might integrate also an internal representation of the previous haptic experience (Stilla and Sathian 2008, Del Vecchio et al. 2018). This point has been now better specified in the Discussion.

7) In Table, the legend states that "no leads were responsive only in the observation condition". This appears to contradict the table itself, as it shows that 1 ipsilateral lead was responsive in the observation only condition during the reach-to-grasp phase. Please ensure this information is reported correctly.

Please, see the updated caption of table 1.

8) The image in Figure 1 appears to be somewhat degraded and unclear, as if it has been photocopied. Is this the intended image?

We apologize for the lack of clarity of the previous version of Figure 1. It has been now updated: it is left in color with blurring effect. Furthermore, to better clarify the procedure we added a timeline detailing each phase of the experimental paradigm.

Finally, a note about your interpretation of the SII responsiveness to median nerve stimulation:

9) I am very satisfied with how you interpret the large differences in SI vs. SII activation in response to this nerve stimulation, and its integration into your

conclusions regarding the different functional properties of SI and SII during action execution. However, the authors may also like to consider higher-level concepts such as the Sense of Agency (i.e., the sense that "I am performing this action"). Assuming that SII is implicated in higher-level control of actions (as you have stated), it may be interesting to compare SII activation during actions that are specifically executed by the patient, and actions that are externally-stimulated. This could either be nerve stimulation, or simply an experimenter physically moving the arm and hand of the patient to grasp an object. Would SI/SII activation be similar during these two types of 'executed' actions, or would an externally-controlled action produce neural responses similar to observing another's action?

This is a very good point. In this study we decided to focus on the role of SII only on active movements. Although previous studies stated the involvement of SII in the execution and observation of actions (e.g. Ishida et al. 2011, Hihara et al. 2015), they did not provide any temporal characterization of the responses for this area. Thanks to intracranial EEG, instead, we got the chance to investigate SII with a 10-ms resolution scale and to disentangle its behavior over the different phases of the experimental paradigm. SII displays a sustained behavior starting just before the hand-object interaction and lasting for the whole manipulative action, both in execution and in observation. However, if we consider the term starting after hand lifting until the hand-object interaction (i.e. the core of the reaching phase), we can observe that there is not significant activation in the observation condition, thus ruling out that SII might mirror an intransitive movement. The sustained responsiveness of the manipulation phase, instead, might indicate that SII rather mirrors movements involving a haptic component. Coming back to your question, we might speculate that observing "externally-controlled" movements does not produce similar neural responses, since the active haptic component is lacking. It is worth noting, however, that further research is needed to better address and clarify these points.

Aside from all of these points, I was very impressed with the manuscript. It is well placed within the literature and fulfills the need to elucidate the function and anatomy of SII, and its results have a clear impact on future models of the lateral grasping network and of action-mirroring.

REVIEWERS' COMMENTS:

Reviewer #1 (Remarks to the Author):

The manuscript has improved since the last review, and the authors have included some more information relevant to the background of the topic and the theoretical motivations for the work. However, I have remaining concerns which I will outline here.

In several parts of the manuscript, the authors still lack sufficient theoretical information or specific findings from the prior literature. However, in the response to the reviews, some information has been provided which I believe would be appropriate to include in the manuscript. For instance, around line 134, when justifying the choice of gamma band, the manuscript still lacks enough content/detail, however the response to reviewers had more: "In fact, it is agreed that gamma rhythm serves as an indicator of neuronal activation reflecting spiking activity (Manning et al. 2009, Vangeneugden et al. 2008) and being highly functionally and spatially specific (Lachaux et al. 2012, Vidal et al. 2010)." This type of detail is appropriate for the manuscript.

In general, the background and lit review still lacks information about what prior researchers have found. Now, the manuscript does acknowledge that such work exists, however, it is only mentioned in passing and with no mention of the actual results and the state of the field in this area (SI and SII during movement--what is known about this?). The lack of sufficient content makes the current manuscript still feel separate from the existing literature, and significantly limits the impact this work can have in the field. Several concepts are introduced with only surface-level mention and a more considered engagement with the ideas can greatly improve the paper. For instance: line 150, a mention of "which features" of the mirror mechanism are encoded by SII. What is meant by this? What are the possible implications? Are there current theories about which features are encoded, and why is it important? One would not expect a lengthy treatise on these questions, but as it is, the authors simply are not giving enough consideration to the motivation of their study.

Likewise, several reviewers expressed a sense of confusion about the novel contributions of this paper, and the authors have responded that they addressed this on lines 534-539. However, the response in the letter to reviewers is much more deep and informative than the manuscript itself, in which only 4 lines of the conclusion are used to summarize the contribution of this paper. This is not enough to sufficiently build the case for what new knowledge is gained through this work.

The statistics which are included in the response to reviewers should be included in the manuscript (or in the supplementary data if there is some limitation on the manuscript). Additionally, effect sizes should be included, as well as a mention of how multiple comparisons are controlled for.

The manuscript would still require significant copyediting throughout, for more fluent English. However, I do note they have reviewed and fixed many copyediting problems that were present before.

The data availability statement does not appear in the manuscript, and there is not clear information about whether data is shared somewhere, or whether there are logistical/other limitations preventing the sharing of the data. The same goes for the analysis code. The authors state they used custom MATLAB scripts but these are not available, nor is there any explanation of their availability. These issues seem to violate the Nature reporting standards and availability of data, as far as I can tell.

Reviewer #2 (Remarks to the Author):

I thank the author for their work on the manuscript. I recommend the present version for publication.

Reviewer #3 (Remarks to the Author):

The major concerns I had with the original submission have been sufficiently addressed in the revision. I believe this is a very interesting paper that will be of broad interest to the field.

Reviewer #4 (Remarks to the Author):

I am satisfied that all of my points in the previous round have been adequately addressed. I thank the authors for making their clarifications and would be happy to recommend this manuscript for publication in Communications Biology.

REVIEWERS' COMMENTS:

Reviewer #1 (Remarks to the Author):

Dear Editor and Reviewers,

We thank you for the positive comments about our manuscript. Please find below the point-by point replies to all the remaining issues, which correspond to the amendments introduced in the text (see yellow hatchings in the revised manuscript). We believe that the manuscript has now fixed all the unsettled issues, and hope that it may be retained suitable for publication in Communications Biology. Finally, all the authors agreed with a transparent peer review, so please publish the peer review.

Best regards,

Maria Del Vecchio

The manuscript has improved since the last review, and the authors have included some more information relevant to the background of the topic and the theoretical motivations for the work. However, I have remaining concerns, which I will outline here.

In several parts of the manuscript, the authors still lack sufficient theoretical information or specific findings from the prior literature. However, in the response to the reviews, some information has been provided which I believe would be appropriate to include in the manuscript. For instance, around line 134, when justifying the choice of gamma band, the manuscript still lacks enough content/detail, however the response to reviewers had more: "In fact, it is agreed that gamma rhythm serves as indicator of neuronal activation reflecting spiking activity (Manning et al. 2009, Vangeneugden et al. 2008) and being highly functionally and spatially specific (Lachaux et al. 2012, Vidal et al. 2010)."

This type of detail is appropriate for the manuscript.

Thanks for your comment. We apologize for having made this point clearer in the previous response to reviewers than in the manuscript itself. We have now expanded the introduction, explaining the rationale underlying the choice of the gamma band as indicator of neural activity. The related literature has been mentioned and referenced as well. Edits can be found at lines 134-138.

Q2: In general, the background and lit review still lacks information about what prior researchers have found. Now, the manuscript does acknowledge that such work exists, however, it is only mentioned in passing and with no mention of the actual results and the state of the field in this area (SI and SII during movement-what is known about

this?). The lack of sufficient content makes the current manuscript still feel separate from the existing literature, and significantly limits the impact this work can have in the field. Several concepts are introduced with only surface-level mention and a more considered engagement with the ideas can greatly improve the paper. For instance: line 150, a mention of "which features" of the mirror mechanism are encoded by SII. What is meant by this? What are the possible implications? Are there current theories about which features are encoded, and why is it important? One would not expect a lengthy treatise on these questions, but as it is, the authors simply are not giving enough consideration to the motivation of their study.

R2: The reviewer is right, in the previous version of the manuscript we had probably overlooked the functional roles that SII may play both in action execution and in action mirroring. For this reason, we added two different paragraphs in the new discussion: the first (lines: 276-291) concerns the previous studies supporting the pivotal role of SII in manipulative actions and haptic exploration. Given the conclusions that we draw from our results, this section should help the reader to more easily catch the functional parallelism of SII between action execution and observation. The second paragraph (lines: 292-309) reports the interpretations that previous studies ascribed to mirroring activity in SII. As the reviewer himself acknowledged, it would have been not reasonable to make a lengthy treatise of these topics, as they would deserve several pages each. However, we hope that these two additions may help in better grounding our study on previous literature, as well as in framing even better our conclusions.

Q3: Likewise, several reviewers expressed a sense of confusion about the novel contributions of this paper, and the authors have responded that they addressed this on lines 534-539. However, the response in the letter to reviewers is much more deep and informative than the manuscript itself, in which only 4 lines of the conclusion are used to summarize the contribution of this paper. This is not enough to sufficiently build the case for what new knowledge is gained through this work.

R3: The reviewer is right. We have modified the Introduction section (lines 139-149) and the Discussion section (lines 318-353) clarifying the novel contribution of our work.

Q4: The statistics which are included in the response to reviewers should be included in the manuscript (or in the supplementary data if there is some limitation on the manuscript). Additionally, effect sizes should be included, as well as a mention of how multiple comparisons are controlled for.

R4: Thank you for raising this point. The table submitted as part of the response to reviewers has been now moved in the Supplementary Materials. For each of the conducted test, we computed the effect size by mean of Cohen's d , and reported them in the table.

Being t-tests computed time-wise, we reported the range for each phase of responsiveness. All p-values are corrected for Bonferroni, considering the number of time bins in each phase of responsiveness.

The manuscript would still require significant copyediting throughout, for more fluent English. However, I do note they have reviewed and fixed many copyediting problems that were present before.

We apologize for that, we went throughout the text and revised it.

The data availability statement does not appear in the manuscript, and there is not clear information about whether data is shared somewhere, or whether there are logistical/other limitations preventing the sharing of the data. The same goes for the analysis code. The authors state they used custom MATLAB scripts but these are not available, nor is there any explanation of their availability. These issues seem to violate the Nature reporting standards and availability of data, as far as I can tell.

Thanks a lot for the comment. Please find in Method Section the code/data availability statement. Furthermore, we now better clarify that these data come from a clinical population who voluntarily took part in our experimental study. For this reason, both data and code are available under reasonable request from the corresponding author.